# Protective Effect of Tomato-Oleoresin Supplementation on Oxidative Injury Recoveries Cardiac Function by Improving β-Adrenergic Response in a Diet-Obesity Induced Model

**DOI:** 10.3390/antiox8090368

**Published:** 2019-09-02

**Authors:** Artur Junio Togneri Ferron, Giancarlo Aldini, Fabiane Valentini Francisqueti-Ferron, Carol Cristina Vágula de Almeida Silva, Silmeia Garcia Zanati Bazan, Jéssica Leite Garcia, Dijon Henrique Salomé de Campos, Luciana Ghiraldeli, Koody Andre Hassemi Kitawara, Alessandra Altomare, Camila Renata Correa, Fernando Moreto, Ana Lucia A. Ferreira

**Affiliations:** 1Medical School, Sao Paulo State University (Unesp), Botucatu 18618-687, Brazil; 2Department of Pharmaceutical Sciences, University of Milan, 20133 Milan, Italy

**Keywords:** high sugar-fat diet, obesity, β-adrenergic system, cardiac dysfunction, lycopene, tomato-oleoresin

## Abstract

The system redox imbalance is one of the pathways related to obesity-related cardiac dysfunction. Lycopene is considered one of the best antioxidants. The aim of this study was to test if the tomato-oleoresin would be able to recovery cardiac function by improving β-adrenergic response due its antioxidant effect. A total of 40 animals were randomly divided into two experimental groups to receive either the control diet (Control, *n* = 20) or a high sugar-fat diet (HSF, *n* = 20) for 20 weeks. Once cardiac dysfunction was detected by echocardiogram in the HSF group, animals were re- divided to begin the treatment with Tomato-oleoresin or vehicle, performing four groups: Control (*n* = 6); (Control + Ly, *n* = 6); HSF (*n* = 6) and (HSF + Ly, *n* = 6). Tomato oleoresin (10 mg lycopene/kg body weight (BW) per day) was given orally every morning for a 10-week period. The analysis included nutritional and plasma biochemical parameters, systolic blood pressure, oxidative parameters in plasma, heart, and cardiac analyses in vivo and in vitro. A comparison among the groups was performed by two-way analysis of variance (ANOVA). Results: The HSF diet was able to induce obesity, insulin-resistance, cardiac dysfunction, and oxidative damage. However, the tomato-oleoresin supplementation improved insulin-resistance, cardiac remodeling, and dysfunction by improving the β-adrenergic response. It is possible to conclude that tomato-oleoresin is able to reduce the oxidative damage by improving the system’s β-adrenergic response, thus recovering cardiac function.

## 1. Introduction 

Clinical studies show that the excessive body fat leads to many cardiac abnormalities, among them, morphologic and functional changes [1,2]. Animal studies have demonstrated myocardial dysfunction in obese rodents fed with hypercaloric diets [3,4,5,6,7]. Although it is evident that many cardiac changes and/or impairments in performance occur due to adipose tissue accumulation [3,8], the responsible mechanisms by which these changes are not clarified. The system redox unbalance, characterized by a high production of reactive species and inefficient antioxidant activity, is one of the pathways associated with the obesity-related cardiac dysfunction [9].

The β-adrenergic system is one of the most important mechanisms responsible for myocardial contraction and relaxation [10,11,12]. However, chronic expositions to reactive species are associated with sustained adrenergic stimulation, resulting in arrhythmias and heart failure [13]. Considering the redox system’s role in the pathogenesis of obesity and cardiac disorders, the use of antioxidants as therapeutic strategies has been tested [14,15].

Lycopene is a carotenoid present in tomato and red fruits and considered a potent antioxidant [16,17,18]. The tomato, and tomato product consumption are one of the Mediterranean diet’s characteristics, which is associated with health benefits [18]. However, there is a lack of studies regarding lycopene dose ingestion in countries with a Mediterranean diet. Moreover, the few studies which bring information about lycopene consumption have a big variability [19] among the results (for example: in Italy the average intake is 7.4 mg per day while in Spain is 1.6mg per day [18]). The effect of lycopene on cardiovascular disease has been evaluated in clinical [20,21] and experimental studies [22]. Although obesity and oxidative stress are able to lead to cardiac dysfunction, no studies have evaluated the cardiac modulation by lycopene due the antioxidant effect. So, this study aimed to test if the tomato-oleoresin would be able to recovery the cardiac function by improving β-adrenergic response due its antioxidant effect.

## 2. Materials and Methods

### 2.1. Animals and Experimental Protocol

In the present study, male Wistar rats (±187 g) were initially divided into two experimental groups to receive control diet (Control, *n* = 20) or high sugar-fat diet (HSF, *n* = 20) for 20 weeks. The diets and water were provided ad libitum. The diet composition has been described in our previous studies [15,23]. All the animals were housed in an environmental controlled room (22 °C ± 3 °C, 12 h light-dark cycle and relative humidity of 60 ± 5%). All of the experiments were performed in accordance with the National Institute of Health’s Guide for the Care and Use of Laboratory Animals and the procedures were approved by the Animal Ethics Committee of Botucatu Medical School (1196/2016).

At week 20 of this study, the cardiac dysfunction was detected by echocardiogram in the HSF group. Thus, the animals were casually divided to begin the treatment with tomato-oleoresin or vehicle, performing four groups: Control (*n* = 6); Control supplemented with lycopene- tomato oleoresin (Control + Ly, *n* = 6); HSF (*n* = 6) and HSF supplemented with lycopene- tomato oleoresin (HSF + Ly, *n* = 6). Tomato oleoresin was mixed with corn oil correspondent to 10 mg lycopene/kg of body weight (BW) per day and given orally every day, in the morning, for a 10-week period [24,25]. To avoid differences in the energy provided, all the groups received the same corn oil amount (about 2 ml/kg BW per day). The supplementation time and dose were chosen based in previous studies from our research group and others from the literature [24,25,26].

### 2.2. Tomato-Oleoresin Preparation

The tomato-oleoresin (Lyc-O-Mato 6% dewaxed; LycoRed Natural Products Industries, Beersheba, Israel) was mixed with corn oil and kept in the dark, at 4 °C, until the moment to be used [27]. The tomato oleoresin-corn oil mixture stayed for 20 min in a water-bath at 54 °C before the animals receive. The total amount of lycopene in each solution was 5mg/ml. Lycopene stability was confirmed by diode-array spectra at 450 nm, as previously described [28]. 

### 2.3. Nutritional Evaluation

Nutritional evaluation included: feed consumption (FC)-daily consumed amount in grams of chow feed; final body weight (BW); caloric intake (CI), calculating according to the following formula for the control group: caloric intake (kcal/day) = feed consumption (g) × dietary energy (3.59 kcal/g). For the HSF group, the caloric intake was calculated as following: water volume consumed (mL) × 0.25 (equivalent to 25% fructose) × 4 (calories per gram of carbohydrate) + caloric intake providing by the chow (feed consumption (g) × dietary energy (4.35 kcal/g).

Feed efficiency (FE) is defined as the ability to convert the caloric intake to body weight. It was calculated according to the formula: FE (%) = BW gain (g)/total caloric intake (kcal) × 100 [15,23]. The adiposity index, considered an obesity marker, was calculated as follow: adiposity index = (total body fat (BF)/final body weight) × 100. BF was evaluated considering the sum of the individual fat pad weights: BF = epididymal fat + retroperitoneal fat + visceral fat.

### 2.4. Metabolic and Hormonal Analysis

The plasma used for the biochemical analysis was collected after 12 h of fasting. The glucose levels were evaluated by a glucometer (Accu-Chek Performa; Roche Diagnostics, Indianapolis, IN, USA). The insulin levels were analyzed by ELISA assay with commercial kits (Millipore) [23]. The HOMA-IR (homeostatic model of insulin resistance), considered an insulin resistance index, was calculated by the following formula: HOMA-IR = [fasting glucose (mmol/L) × fasting insulin (µU/mL)]/22.5 [15].

### 2.5. Systolic Blood Pressure (SBP)

SBP was evaluated by a non-invasive tail-cuff method with a NarcoBioSystems^®^ Electro-Sphygmomanometer (International Biomedical, Austin, TX, USA) with the conscious rats. For this, the animals were heat during 4–5min in a wooden box (50 × 40 cm), with two incandescent lamps and temperature between 38–40 °C, to induce arterial vasodilation in the tail. Then, the rats were transferred to an iron cylindrical support specially made to allow the total exposure of the animal’s tail [29]. After this procedure, a cuff with a pneumatic pulse sensor was attached to the tail and inflated to 200 mmHg pressure and successively deflated. Blood pressure values were documented on a Gould RS 3200 polygraph (Gould Instrumental Valley View, Cleveland, OH, USA). The final SBP of each animal considered the average of three pressure readings.

### 2.6. Lycopene Bioavailability Evaluation

The presence of lycopene was determined in plasma and cardiac tissue homogenate. To extract the carotenoids, samples were incubated with internal standard (equinenone), chloroform/methanol CHCl_3_/CH_3_OH (3 mL, 2:1, *v*/*v*) and 500 mL of saline 8.5 g/L. Then the samples were centrifuged at 2000× *g* for 10 min and the supernatant was collected and hexane was added. The chloroform and hexane layers were evaporated under nitrogen and the residue was resuspended in 150 mL of ethanol and sonicated for 30 s. 50 μL of this aliquot was injected into the HPLC. The HPLC system was a Waters Alliance 2695 (Waters, Wilmington, MA, USA) and consisted of pump and chromatography bound to a 2996 programmable photodiode array detector, a C30 carotenoid column (5 µm, 150 × 4.6 mm, YMC-Yamamura Chemical Research, Wilmington, NC, USA), and Empower software (Empower 3, chromatographic data software Milford, MA, USA). The HPLC system programmable photodiode array detector was set at 450 nm for carotenoids. The mobile phase consisted of ethanol/methanol/methyl-tert-butyl ether/water (83:15:2, *v*/*v*/*v*, 15 g/L with ammonium acetate in water, solvent A) and methanol/methyl-tert-butyl ether/water (8:90:2, *v/v/v*, 10 g/L with ammonium acetate in water, solvent B). The gradient procedure, at a flow rate of 1 mL/min (16 °C), was as follows: (1) 100% solvent A was used for 2 min followed by a 6 min linear gradient to 70% solvent A; (2) a 3 min hold followed by a 10 min linear gradient to 45% solvent A; (3) a 2 min hold, then a 10 min linear gradient to 5% solvent A; (4) a 4 min hold, then a 2 min linear gradient back to 100% solvent A. For the quantification of the chromatograms, a comparison was made between the area ratio of the substance and area of the internal standard obtained in the analysis [30].

### 2.7. Cardiac Malondialdehyde (MDA) Levels

MDA is the main lipid peroxidation marker. It is considered an oxidative stress index [31] and associated with cardiovascular diseases [32]. Thus, MDA levels were used to evaluate the cardiac lipid oxidation as follow: 

Cardiac tissue (±150 mg) was homogenized (ULTRA-TURRAX® T25 basic IKA^®^ Werke Staufen/Germany) with 1.0 mL of cold phosphate buffered saline (PBS) pH 7.4, and centrifuged at 800 g at 4 °C for 10 min. Then, 100 µL from the supernatant was mixed with 700 μL of 1% orthophosphoric acid and 200 μL of thiobarbituric acid (42 mM). After this, the samples were kept at 100 °C for 60 min in a water bath, and immediately cooled on ice. In a 2 mL tube, 200 μL was mixed with 200 μL sodium hydroxide/methanol (1:12 *v/v*). After vortex, the samples were centrifuged for 3 min at 13,000 *g*. 200 μL from the supernatant was transferred to a glass vial and 50μL was injected into the column. The HPLC used was a Shimadzu LC-10AD system (Kyoto, Japan) with a C18 Luna column (5 μm, 150 × 4.60 mm, Phenomenex Inc., Torrance, CA, USA), and a Shimadzu RF-535 fluorescence detector (excitation 525 nm, emission 551 nm), and 0.5mL/min phosphate buffer flow (KH2PO4 1mM, pH 6.8) [25]. The MDA levels considered the peak area determination in the chromatograms relative to the standard curve of known concentrations.

### 2.8. Circulating Advanced Oxidation Protein Products 

Advanced oxidation protein products (AOPPs) are oxidized plasma proteins resulting from the exposure to oxidation products and are transported by albumin in the circulation [33]. The literature reports that high AOPP circulating levels contribute to cardiac diseases [34]. 

AOPP determination was based on spectrophotometric detection according to Kalousova et al. [35]. Plasma samples (200 µL) were diluted 1:5 with PBS. It was also used 200 µL of chloramin T (0–100 µmol/L) for calibration curve and the blank was only PBS (200 µL). All the samples were put on a microtiter plate and mixed with 10 µL of KI 1.16 M and 20 µL of acetic acid. The absorbance was measured immediately at 340 nm (spectrophotometer Multiskan Ascent, Labsystems, Vantaa, Finland). The final AOPP concentration is expressed in chloramine units (µmol/L).

### 2.9. Circulating Carboxymethyl Lysine 

Advanced glycation end products (AGEs) are a group of several molecules generated by both non-enzymatic glycation and protein, lipids and nucleic acids oxidation, able to modify tissue function and mechanical properties [36]. In vivo, CML is the main AGE associated with cardiac pathologies [37]. The plasmatic carboxymethyl lysine (CML) levels were evaluated using an ELISA commercial kit (OxiSelect™ CML, Cell Biolabs Inc., San Diego, CA, USA) following the manufacturer’s instructions. 

### 2.10. Echocardiographic Study 

The analyze was performed in the live animals by transthoracic echocardiography, with a Vivid S6 system equipped with multifrequency ultrasonic transducer 5.0 to 11.5 MHz (General Electric Medical Systems, Tirat Carmel, Israel). The animals were lightly anesthetized by intraperitoneal injection with a mixture of ketamine (50 mg/kg) and xylazine (1 mg/kg), put in left decubitus position and only one examiner made all the exams. The heart image structural measurements were obtained in one-dimensional mode (M-mode) guided by the images in two-dimensional mode with the transducer in the parasternal position, minor axis. Left ventricular (LV) evaluation was performed with the cursor M-mode just below the mitral valve plane at the level of the papillary muscles. The aorta and left atrium images were obtained by positioning the M-mode course to plan the aortic valve level [23].

The following cardiac structures were evaluated: diastolic diameter (LVDD); systolic (LVSD) LV; left ventricle diastolic thickness posterior wall (LVPWD); aorta diameter (AD); left atrium (LA). The LV diastolic function was assessed by the transmitral flow early peak velocity (E). The LV systolic function was evaluated by ejection fraction and posterior wall shortening velocity (PWSV). The joint assessment of diastolic and systolic LV function was performed using the Tei index (sum of isovolumetric contraction and IRT time, divided by the left ventricular ejection time). The study was complemented by tissue Doppler evaluation, considering early diastolic (E’), and late (A’) of the mitral annulus (arithmetic average travel speeds of lateral and septal walls), and the ratio by the waves E and E’ (E/E’).

### 2.11. Myocardial Function by Isolated Papillary Muscle Study

Besides echocardiographic analysis, myocardial function was also assessed by LV isolated papillary muscles. This procedure has been used by several authors [6,7,29]]. Conventional mechanical parameters at *Lmax* were calculated from isometric contraction: maximum developed tension normalized per cross-sectional area (DT [g/mm^2^]), resting tension normalized per cross-sectional area (RT [g/mm^2^]), positive (+d*T*/d*t* [g/mm^2^/s]) and negative (−d*T*/d*t* [g/mm^2^/s]) tension derivative normalized per cross-sectional area of papillary muscle (CSA).

### 2.12. β-Adrenergic System Study

β-adrenergic receptors (βAR) are important to regulate cardiac function in both normal and pathologic conditions [11]. The receptors activity was assessed by the dose-response relationship between the isoproterenol and conventional mechanical parameters of papillary muscle at *Lmax*. After baseline values determination, the isoproterenol was added to the vat in the presence of 1.0 mM [Ca^2+^] to increase progressively the concentrations for 10^−8^, 10^−7^ and 10^−6^ mol/L. 

The stabilization of contractile response occurs nearly 3–5min after adding each isoproterenol dose. Data were sampled and expressed as the stimulation mean percent (%) [29]. At the end of the study, length (mm), weight (mg), and CSA (mm^2^) [38] were measured for papillary muscle characterization. The CSA was calculated from papillary muscle length and weight, assuming uniformity and a specific gravity of 1.0. The muscle length at *Lmax* was measured with a cathetometer (Gartner Scientific Corporation, Chicago, IL, USA), and the muscle between the two clips was blotted dry and weighed.

### 2.13. Statistical Analysis 

The results are expressed in mean ± standard deviation (SD). Two-way analysis of variance (ANOVA) for independent samples was used to determine the differences among the groups. In order to evaluate the positive and negative inotropic effects on myocardial function, it was used a repeated-measures two-way ANOVA. Once detected significant differences (*p* < 0.05), the Tukey post hoc test for multiple comparisons were carried out. All the statistical analyses were performed using SigmaStat for Windows (Version 3.5, San Jose, CA, USA). 

## 3. Results

The lycopene bioavailability is presented in the Table 1. It is possible to verify the presence of lycopene in both groups, which were supplemented (Control + Ly and HSF + Ly).

The HSF group presented increased caloric intake (kcal/d), final body weight (g), adiposity index, glucose levels, HOMA-IR and systolic blood pressure values compared to the control group. The HSF + Ly showed the same changes observed in HSF group when compared to control + Ly, except for HOMA-IR. Tomato-oleoresin suppressed the insulin resistance in HSF + Ly compared to HSF (Figure 1). No effect was observed of tomato-oleoresin on the other parameters. 

The HSF group presented cardiac remodeling (increased LVDS, LVPWD and reduced LVDD), and deterioration of both systolic (decreased ejection fraction, Tei-a and Tei-b) and diastolic (increased E/E′ and decreased Tei-a and Tei-b) functions compared to control group. Regarding the tomato-oleoresin supplementation effect, HSF + Ly group showed improvement in some remodeling, systolic and diastolic parameters compared to HSF (Table 2).

The myocardial papillary muscle study at baseline condition with 2.5 mM Ca^2+^ is presented in the Table 3. HSF group showed functional impairment in the maximum developed tension (DT) compared to control group. Tomato-oleoresin supplementation was effective to recovery the DT capacity in HSF + Ly group compared to HSF (Table 3). 

Figure 2 shows the β-adrenergic stimulation on the papillary muscle function. The isoproterenol stimulation demonstrated that the HSF group presented functional impairment in DT (10^−6^ M) and −d*T*/d*t* (10^−7^ and 10^−6^ M) compared to control group. Tomato-oleoresin supplementation was effective to recover the −d*T*/d*t* (10^−7^ and 10^−6^ M) capacity in HSF + Ly group compared to HSF.

Figure 3 shows the oxidative stress parameters in plasma and cardiac tissue. All the parameters increased in HSF group compared to control group. By contrast, it is possible to note that CML, AOPP and cardiac MDA plasma levels reduced in HSF + Ly group in respect to HSF to demonstrate a positive effect of tomato-oleoresin. 

## 4. Discussion

This study aimed to test if the tomato-oleoresin would be able to recovery the cardiac function by improving *β*-adrenergic response due its antioxidant effect. The results show that the HSF groups presented with obesity, characterized by the higher values of body weight and adiposity index, and metabolic syndrome, with insulin resistance, dyslipidemia, and hypertension, all diseases usually associated with obesity [39]. These findings confirm that the diet model used in this study was efficient to lead obesity and related disorders, corroborating the literature [6,7,15,23]. Regarding the lycopene effect on obesity and related disorders, it was observed a positive action only on insulin resistance in the HSF + Ly group, represented by the reduction in HOMA-IR. The literature attributes the tomato-oleoresin benefic effects on diabetes to the lycopene antioxidant potential [40]. Another explanation for this amelioration is the anti-inflammatory effect of tomato-oleoresin. Since insulin resistance and type 2 diabetes are conditions closely related with inflammation and studies already showed that tomato-oleoresin ameliorates the inflammation, this property may explain the beneficial effect on glucose metabolism [41]. The antioxidant and anti-inflammatory effect of lycopene can also be explained by considering its well-established ability, through electrophilic metabolites, to activate Nrf2 pathway thus inducing phase II detoxifying/antioxidant enzymes and inhibiting NF-κB activation [42,43,44].

Obesity is also associated with cardiac abnormalities, among them morphological, hemodynamic and functional alterations [1,8,23]. Considering the lycopene absence effect on obesity and hypertension in the HSF + Ly group, should both HSF groups present cardiac damage. However, the echocardiographic analysis showed cardiac remodeling and impairment in ventricular systolic and diastolic function only in HSF group after 30 weeks. In opposition, the HSF group supplemented with tomato-oleoresin showed a cardiac remodeling and function recovery. 

Several mechanisms could explain the obesity-induced cardiac dysfunction, among them is the β-adrenergic system responsiveness. The myocardial β-adrenergic mechanism is the main responsible by regulating the cardiac performance, especially by intracellular Ca^2+^ handling [6,7]. Although functional studies using isolated papillary muscle have showed that obesity is able to lead to impairment in cardiac contractile [3,5,7], a small number of studies have evaluated the β-adrenergic response in high sugar-fat diet obesity-induced experimental models [29,45,46,47,48,49]. Our results demonstrated that the isoproterenol stimulation leaded to negative responses in both systolic (DT) and diastolic (−d*T*/d*t*) response in the HSF group while the HSF + Ly group showed an improvement in the β-adrenergic response. However, it is still unclear how the high sugar-fat diet obesity-induced leads to a reduction in the β-adrenergic response.

One hypothesis for the β-adrenergic response impairment is the chronic exposition to reactive oxygen species (ROS) promoted by obesity [9,50]. The literature reports that the direct contact with ROS exerts the same action of isoproterenol on β-adrenergic response, increasing the calcium transient amplitude, therefore, exerting a modulator role in the myocardial contractility [13]. However, this continues exposition to ROS may result in deleterious effects and contribute the development of cardiac arrhythmias and failure [9]. Considering the lycopene antioxidant effect, the amelioration in the β-adrenergic responsiveness of the HSF + Ly group can be attributed to this carotenoid property [14]. 

Another hypothesis is that the redox system imbalance in obesity conditions may lead to damage to lipids and proteins, generating such biomarkers as MDA, CML and AOPP, which were evaluated in this study [33,51,52,53]. These oxidative products can damage directly the cardiac tissue by altering its geometry and functionality, or indirectly by the carbonylation of proteins involved in the myocardial contractility regulatory response, as the *β*-adrenergic pathway [13,54,55]. While the HSF group presented higher levels of MDA, CML and AOPP and cardiac function deterioration, the tomato-oleoresin antioxidant effect is confirmed by reduced levels of these markers and cardiac function recovery in the HSF + Ly group. 

## 5. Conclusions

In summary, this study found that the HSF diet induced obesity-related cardiac dysfunction and the tomato-oleoresin was able to attenuate this condition. Therefore, it is possible to conclude that tomato-oleoresin is able to reduce oxidative damage, thereby improving the system’s β-adrenergic response and recovering cardiac function.

## Figures and Tables

**Figure 1 antioxidants-08-00368-f001:**
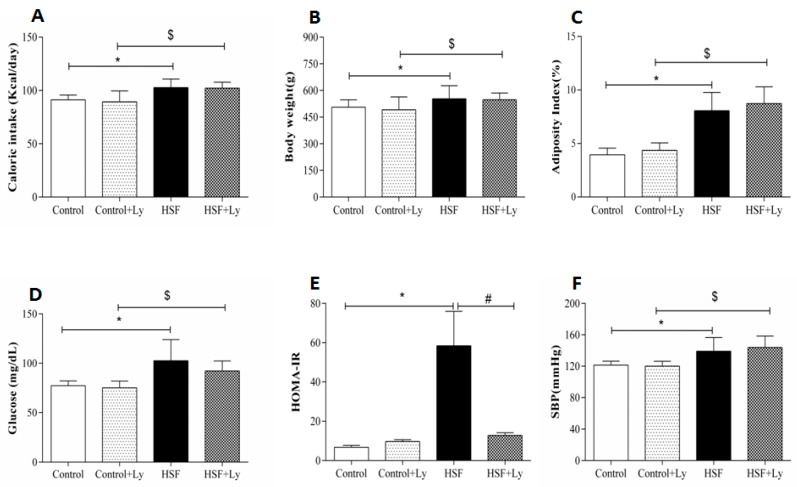
Nutritional and cardio- metabolic parameters. **A**—caloric intake (kcal/day); **B**—adiposity index (%); **C**—final body weight (g); **D**—glucose (mg/dL); **E**—HOMA-IR; **F**—systolic blood pressure (mmHg). Data are expressed in mean ± standard deviation (*n* = 6 animals/group). Comparison by Two-way ANOVA with Tukey post-hoc (*p* < 0.05): * HSF *vs* Control; ^#^ HSF *vs* HSF + Ly; ^$^ HSF + Ly *vs* Control + Ly.

**Figure 2 antioxidants-08-00368-f002:**
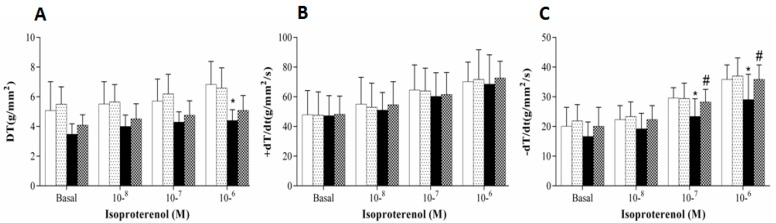
β-adrenergic stimulation in papillary muscles. Data are expressed in mean ± standard deviation (*n* = 6 animals/group). Baseline calcium concentration (1.0 mM) is presented as 100%. A, Maximum developed tension normalized per cross-sectional area [DT, g/mm^2^]. B, positive [+d*T*/d*t*, g/mm^2^/s] and C, negative [−d*T*/d*t*, g/mm^2^/s] tension derivative normalized per cross-sectional area of the papillary muscle. Two-way ANOVA repeated-measures with Tukey post-hoc was used to compare the groups (*p* < 0.05); * HSF *vs* Control; ^#^ HSF *vs* HSF + Ly.

**Figure 3 antioxidants-08-00368-f003:**
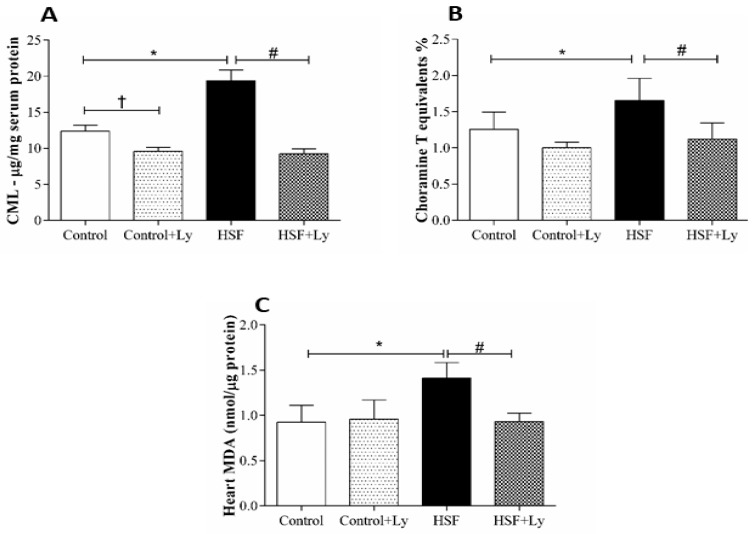
Plasma and cardiac tissue redox state parameters. **A**—Carboxymethyl lysine (CML-pg/mg protein); **B**—Cholaramine T equivalents %; **C**—Malondyhaldeide (MDA-nmol/μg protein). Data are expressed in mean ± standard deviation (*n* = 6 animals/group). Comparison by Two-way ANOVA with Tukey post-hoc (*p* < 0.05), * HSF *vs* Control; ^#^ HSF *vs* HSF + Ly.

**Table 1 antioxidants-08-00368-t001:** Lycopene Bioavailability.

Lycopene Concentration	Groups			
	Control	Control + Ly	HSF	HSF + Ly
Plasma (µg/mL)	ND	3.61 ± 0.68	ND	3.59 ± 2.31
Heart (µg/g of tissue)	ND	4.83 ± 2.37	ND	2.41 ± 0.36

Data are expressed in mean ± standard deviation (*n* = 4 animals/group). ND: Not detectable.

**Table 2 antioxidants-08-00368-t002:** Echocardiographic study.

Variables	Groups				Effect		
	Control	Control + Ly	HSF	HSF + Ly	Diet	Ly	Interaction
**LVDD (mm)**	7.15 ± 0.11	7.02 ± 0.11	6.70 ± 0.12 *	6.92 ± 0.11	0.019	0.665	0.123
**LVDS (mm)**	2.83 ± 0.10	2.74 ± 0.10	3.17 ± 0.11 *	2.91 ± 0.10	0.016	0.098	0.417
**LVPWD (mm)**	1.63 ± 0.04	1.53 ± 0.04	1.73 ± 0.04 *	1.62 ± 0.04 ^#^	0.031	0.014	0.932
**AD (mm)**	3.91 ± 0.06	3.86 ± 0.06	3.89 ± 0.07	3.88 ± 0.06	0.999	0.682	0.740
**LA (mm)**	4.86 ± 0.11	4.85 ± 0.11	5.02 ± 0.11	4.88 ± 0.11	0.388	0.483	0.536
**HR (bpm)**	254 ± 14	265 ± 14	262 ± 15	262 ± 14	0.871	0.716	0.713
**E (cm/s)**	73.6 ± 2.1	73.2 ± 2.18	76.1 ± 2.1	75.1 ± 2.3	0.351	0.742	0.895
**PWSV (cm/s)**	58.6 ± 1.3	61.1 ± 1.3	56.1 ± 1.3	59.8 ± 1.4	0.181	0.028	0.622
**Dec. time (ms)**	47.2 ± 1.3	42.1 ± 1.3	50.6 ± 1.3	42.7 ± 1.4	0.128	<0.001	0.322
**Tei-a (ms)**	116.1 ± 2.5	116.8 ± 2.5	99.1 ± 2.5 *	111.7 ± 2.6 ^#^	<0.001	0.012	0.024
**Tei-b (ms)**	86.6 ± 2.9	92.6 ± 2.9	77.7 ± 2.9 *	85.5 ± 3.1 ^#^	0.012	0.028	0.761
**EF (%)**	0.93 ± 0.008	0.93 ± 0.008	0.88 ± 0.008 *	0.93 ± 0.008 ^#^	<0.001	0.006	0.008
**E/E’**	13.3 ± 0.4	12.7 ± 0.4	15.3 ± 0.4 *	13.9 ± 0.50 ^#^	0.002	0.049	0.439

Data are expressed in mean ± standard deviation (*n* = 6 animals/group). Comparison by Two-way ANOVA with Tukey post-hoc (*p* < 0.05): * HSF *vs* Control; ^#^ HSF *vs* HSF+Ly. LVDD, left ventricular diastolic diameter; LVSD, left ventricular systolic diameter; LVPWD, diastolic thickness posterior wall of the left ventricle; AD, aorta diameter; LA, left atrium diameter during ventricular systole; HR, heart rate; E, E-wave peak transmitral early diastolic inflow velocity; PWSV, posterior wall shortening velocity; Dec. time, deceleration time; Transmitral flow, Tei-a and Tei-b; EF, ejection fraction; E/E’.

**Table 3 antioxidants-08-00368-t003:** Isolated papillary muscle at baseline condition (2.5 mM Ca^2+^).

Variables	Groups				Effect		
	Control	Control + Ly	HSF	HSF + Ly	Diet	Ly	Interaction
**DT(g/mm^2^)**	5.96 ± 1.25	6.29 ± 1.65	4.41 ± 1.11 *	6.05 ± 1.19 ^#^	0.066	0.046	0.173
**RT(g/mm^2^)**	0.65 ± 0.11	0.61 ± 0.11	0.63 ± 0.08	0.57 ± 0.11	0.512	0.202	0.844
**+d*T*/d*t*(g/mm^2^/s)**	61.9 ± 10.1	63.5 ± 18.4	60.8 ± 11.7	65.5 ± 19.7	0.934	0.573	0.773
**−d*T*/d*t*(g/mm^2^/s)**	16.8 ± 2.4	17.5 ± 2.9	15.5 ± 3.3	16.1 ± 2.9	0.193	0.569	0.933
**CSA(mm^2^)**	1.11 ± 0.12	1.10 ± 0.23	1.25 ± 0.27	1.17 ± 0.3	0.181	0.801	0.912

Data are expressed in mean ± standard deviation (*n* = 6 animals/group). Comparison by Two-way ANOVA with Tukey post-hoc (*p* < 0.05): * HSF *vs* Control; ^#^ HSF *vs* HSF+Ly. DT, Maximum developed tension normalized per cross-sectional area of the papillary muscle; RT, Resting tension normalized per cross-sectional area of the papillary muscle; peak of the positive, +d*T*/d*t* and negative, −d*T*/d*t* tension derivatives normalized per cross-sectional area of the papillary muscle; CSA, cross-sectional area.

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
