# Peer review of "Protective Effect of Tomato-Oleoresin Supplementation on Oxidative Injury Recoveries Cardiac Function by Improving β-Adrenergic Response in a Diet-Obesity Induced Model"

_antioxidants, 2019, doi:10.3390/antiox8090368_

Round 1

Reviewer 1 Report

This is a very nice paper. I suggest to describe the correspondence between the dose of licopene used in the experiments and that generally contained in the Mediterranean diet.

Moreover as licopene affects nadph oxidase activity it would be of interest to dose in your model a marker of nadph oxidase activity.

Reviewer 2 Report

The study is well designed and executed.

Author Response

Dear Reviewr, Thank you very much for your consideration.

Reviewer 3 Report

The authors of this paper test the effect of tomato-oleoresin on cardiac function and they investigate if its supplementation and its antioxidant propriety improve β- adrenergic response.

Their results show that high sugar-fat diet (HSF) induced obesity-related cardiac dysfunction and the tomato-oleoresin was able to attenuate this condition. Therefore, they conclude that tomato-oleoresin is able to reduce the oxidative damage improving the system β-adrenergic response recovering the cardiac function.

I think the work has many points to clarify, in particular:

1.     The authors add to the supplementation of tomato-oleoresin the corn oil. How did the authors exclude the contribution of corn oil?

2.     The authors, how they chose the daily dose and the timing of the study (10 weeks of supplementation)?

3.     The tomato-oleoresin is administered orally, did the authors evaluate its bioavailability?

4.     The authors affirm that tomato oleoresin improves the inflammation and this property may explain the beneficial effect on glucose metabolism. To better investigate this issue, they should evaluate the anti-inflammatory effect of tomato-oleoresin in own model of study by evaluation of pro- and anti-inflammatory cytokines (e.g.  TNF-alpha, IL-4).

5.     PKC plays an important role in intracellular Ca2+ signaling. Moreover, PKC is involved in the activation of pro-oxidant mechanisms. I suggest investigating this pathway to provide a hypothesis on the molecular mechanism involved.

Round 2

Reviewer 1 Report

the authors only partially answeren my questions

Reviewer 3 Report

Thanks for your answers. I have no further comments.